



# The significant role of biomass burning aerosols in clouds and radiation in the South-eastern Atlantic Ocean

Haochi Che[1, a], Philip Stier[1], Hamish Gordon[2, b], Duncan Watson-Parris[1], Lucia Deaconu[1]

[1]Atmospheric, Oceanic and Planetary Physics, Department of Physics, University of Oxford, Oxford, OX1 3PU, UK
[2]School of Earth and Environment, University of Leeds, LS2 9JT, UK
[a]now at: Department of Geophysics, Tel-Aviv University, 69978, Israel
[b]now at: Engineering Research Accelerator, Carnegie Mellon University, Pittsburgh, PA 15217, United States

*Correspondence to*: Haochi Che (haochiche@tauex.tau.ac.il)

**Abstract**

The South-eastern Atlantic Ocean (SEA) is semi-permanently covered by one of the most extensive stratocumulus cloud decks on the planet and experiences about one-third of the global biomass burning emissions from the southern Africa savannah region during the fire season. To get a better understanding of the impact of these biomass burning aerosols on clouds and radiation balance over the SEA, the latest generation of the UK Earth System Model (UKESM1) is employed. Measurements

from the CLARIFY and ORACLES flight campaigns are used to evaluate the model, demonstrating that the model has good skill in reproducing the biomass burning plume. To investigate the underlying mechanisms in detail, the effects of biomass burning aerosols on the clouds are decomposed into radiative effects (via absorption and scattering) and microphysical effects (via perturbation of cloud condensation nuclei (CCN) and cloud microphysical processes). The July-August means are used to characterise aerosols, clouds and the radiation balance during the fire season. Results show around 68% of CCN at 0.2%

supersaturation in the SEA domain can be attributed to biomass burning. The absorption effect of biomass burning aerosols is the most significant in affecting clouds and radiation. Near the continent it increases the maximum supersaturation diagnosed by the activation scheme, while further from the continent it reduces the altitude of the maximum supersaturation. As a result, the cloud droplet number concentration shows a similar pattern. The microphysical effect of biomass burning aerosols decreases the maximum supersaturation and increases the cloud droplets concentration over the ocean; however, this change

is relatively small. The liquid water path is also significantly increased over the SEA (mainly caused by the absorption effect of biomass burning aerosols) when biomass burning aerosols are above the stratocumulus cloud deck. The microphysical pathways lead to a slight increase in the liquid water path over the ocean. These changes in cloud properties indicate the significant role of biomass burning aerosols on clouds in this region. Among the effects of biomass burning aerosols on radiation balance, the semi-direct radiative effects (rapid adjustments induced by biomass burning aerosols radiative effects)

have a dominant cooling impact over the SEA, which offset the warming direct radiative effect (radiative forcing from biomass burning aerosol–radiation interactions). However, the magnitude and the sign of the semi-direct effects are dependent on the relative location of biomass burning aerosols and clouds. The net biomass burning aerosols radiative effect shows a negative



cooling effect in the SEA, indicating the significant role of biomass burning aerosols in affecting the regional radiation balance and climate.

## 1 Introduction

The South-eastern Atlantic Ocean (SEA) is covered semi-permanently by one of the most extensive stratocumulus cloud decks on the planet (Wood, 2012). These clouds reflect a significant amount of solar radiation. Hence, even a moderate change in the cloud deck coverage (15-20 % increase) or liquid water path (20-30 % increase) would produce a negative radiative effect that could completely compensate the radiative forcing of greenhouse gases (Wood, 2012). From July through October, the widespread biomass burning across the savannah region in southern Africa contributes about one-third of the global biomass
burning emissions (Roberts et al., 2009; van der Werf et al., 2010). The emitted biomass burning aerosols (BBA) in southern Africa are transported over the SEA, resulting in different impacts on the underlying stratocumulus deck and radiative balance through multiple interactions (Adebiyi and Zuidema, 2016; Wilcox, 2012; Wood, 2012).

BBA can warm the lower troposphere and modify the radiation budget as they absorb shortwave radiation. At the top of
atmosphere, BBA can exert either a cooling or a warming shortwave direct radiative effect (radiative forcing from BBA–radiation interactions) depending on the underlying layer brightness (e.g., ocean or stratocumulus cloud deck) (Chand et al., 2009; Wilcox, 2012). Despite the fact that intensive studies have been performed (Chand et al., 2009; Lu et al., 2018; Sakaeda et al., 2011; Stier et al., 2013; Wilcox, 2012), there is still no consensus on the magnitude or even the sign of the BBA direct radiative effect over the SEA. This discrepancy is primarily owing to the uncertainties in the underlying cloud coverage (Stier
et al., 2013) and the BBA spatial distribution; therefore, accurate modelling of the spatial and vertical distribution of the BBA plume and clouds is a critical task in this area.

The interactions between BBA and the underlying cloud deck adds additional complication as BBA can alter the thermodynamic structure of the atmosphere (through rapid adjustments induced by BBA radiative effects, i.e., semi-direct
effects) and also serve as additional cloud condensation nuclei (CCN). The former is referred to as BBA radiative effect on cloud, and the latter is BBA microphysical effect on cloud. Both effects have a significant impact on the cloud liquid water path (LWP), cloud coverage, and radiation balance (Gordon et al., 2018; Lu et al., 2018; Wilcox, 2010). When the BBA layer is above the cloud deck, its radiative effect can enhance the existing temperature inversion and therefore stability, inhibiting cloud-top entrainment. As a consequence, boundary layer relative humidity is preserved and cloud coverage maintained. This
could lead to an increase of LWP, optically thicker clouds, and therefore an additional cooling semi-direct effect - potentially of comparable magnitude to the warming BBA direct radiative effect, resulting in both the sign and the magnitude of the total BBA radiative effect remaining unclear (Deaconu et al., 2019; Sakaeda et al., 2011; Wilcox, 2010, 2012). Previous efforts



mainly focused on the above cloud BBA radiative effect, as the BBA plume is generally well separated from the underlying cloud deck in their experiments (Hobbs, 2002; Wilcox, 2012). However, recent studies found that parts of the plume enters the marine boundary layer (MBL) and interacts with clouds (LeBlanc et al., 2019; Lu et al., 2018). These findings are also supported by the possible BBA effects on changing cloud properties from satellite observations (Costantino and Bréon, 2010,

2013; Painemal et al., 2014). Moreover, studies have suggested that as the MBL deepens further offshore, most BBA subsides and are entrained into the MBL (Costantino and Bréon, 2010, 2013; Gordon et al., 2018; Painemal et al., 2014). When the BBA plume enters and interacts with clouds, the microphysical effect of BBA is non-negligible, as BBA can serve as CCN, become activated, and increase the CDNC, resulting in optically thicker clouds of higher albedo (Twomey, 1974, 1977). However, some studies have found that when the LWP remains constant, the increased CDNC will increase cloud-top

entrainment by the fast evaporation of small droplets at the cloud top, which, in return, can reduce cloud fraction and LWP (Wood, 2012). As a result, the BBA microphysical effect on clouds may be diminished or even cancelled out under some scenarios (Ackerman et al., 2004; Wood, 2007). A recent study found the BBA number concentration and hygroscopicity played different roles in modulating CDNC concentration in clean and polluted environments (Kacarab et al., 2020), addding more uncertainty of the BBA microphysical effect. As to the BBA radiative effects, when BBA enter the clouds, it can *"burn*

*off"* clouds by absorbing shortwave solar radiation, warming the air and the accompanying increase in saturation vapour pressure (Hansen et al., 1997; Hill et al., 2008; Koch and Genio, 2010), which can lead to a decrease of both the cloud LWP and the cloud coverage. Therefore, BBA microphysical and radiative effects can play an opposing role for cloud physical and radiative properties, creating significant uncertainties in the net effective radiative forcing (change in net downward radiative flux at the top of the atmosphere after allowing rapid adjustments) associated with BBA in the SEA area. Hence, it is critical

to assess the BBA effects over the SEA during the fire season using a model that can account for all the relevant processes.

The complex interactions between cloud microphysics, radiation, cloud entrainment processes and in particular, the small spatial scales involved make the simulation of the stratocumulus clouds deck in the SEA a challenge. Hence, related process studies mainly rely on high-resolution limited-area models (Gordon et al., 2018; Lu et al., 2018). However, ultimately it is

important to represent and constrain the related effects in General Circulation Models (GCM) widely used to investigate climate responses to anthropogenic perturbations, e.g. by the Intergovernmental Panel on Climate Change (IPCC). Furthermore, to have a better understanding of the simulation and errors of the stratocumulus cloud deck and BBA layer over the SEA in current climate models, the GCM is a necessary tool. To evaluate the model performance, we use two flight campaigns that took place in the SEA to compare with the model simulation. One is the ORACLES (Observations of Aerosols above Clouds

and their interactions) campaign (Redemann et al., in preparation) including three deployments, which were conducted from Namibia in 2016 and from São Tomé in 2017, 2018 (not used), ranging from the west coast of Africa to Ascension Island. The other is the CLARIFY (Clouds and Aerosol Radiative Impacts and Forcing: Year 2016) campaign (Haywood et al., in preparation), which was conducted from Ascension Island in 2017, and located around the Ascension Island. These flight campaigns were carried out during the biomass burning seasons, and are able to provide an ideal dataset covering both BBA



above and interacting with clouds, as previous studies have found that the BBA plume layer generally subsides and meets the gradually deepening marine boundary layer in the vicinity of Ascension Island and St Helena (Adebiyi et al., 2015). In this paper, we combine simulations using the UK Earth System Model (UKESM1) with CLARIFY and ORACLES aircraft campaigns to decompose the effect of the BBA plume into radiative effects and the microphysical effects, and ultimately
investigate the effective radiative forcing associated with aerosol-cloud interactions in the SEA.

## 2 Method

The first version of the United Kingdom Earth System Model, UKESM1(Sellar et al., 2019) is the latest Earth system model developed jointly by the UK's Met Office and Natural Environment Research Council (NERC). The core of UKESM1 is based on the Hadley Centre Global Environmental Model version 3 (HadGEM3) Global Coupled (GC) climate configuration of the
Unified Model (UM) (Hewitt et al., 2011), comprised of the UM atmosphere (Walters et al., 2017), ocean (Storkey et al., 2018), land surface and sea ice components (Ridley et al., 2018; Walters et al., 2017). Aerosol and its interaction with cloud are represented by the UK Chemistry and Aerosol model (UKCA) (Mulcahy et al., 2020; O'Connor et al., 2014), including the modal aerosol microphysics scheme GLOMAP (Mann et al., 2010), with five interactive log-normal aerosol modes comprised of sulfate, sea salt, black carbon, and organic carbon chemical components. Mineral dust is represented separately by an
externally mixed bin representation (Woodward, 2001).

For BBA emissions, we use the global fire assimilation system (GFAS) version 1 data. GFAS is based on satellite fire radiative power (FRP) products and has been operating in real-time under Monitoring Atmospheric Composition and Change (MACC) project (Kaiser, J.W. et al., 2012). The GFAS biomass burning emissions are scaled by 2.0 to improve the agreement with
observations, as suggested in the model configuration (Johnson et al., 2016), with scale factors commonly used for this emission inventory (Kaiser, J.W. et al., 2012). For other emissions, the Coupled Model Intercomparison Project Phase 6 (CMIP6) emission data during 2014 are used (Eyring et al., 2016; Gidden et al., 2019).

The model is configured as Global Atmosphere 7.1 (GA7.1), and our simulations run with a horizontal resolution of N96, i.e.,
1.875º × 1.25º, and 85 vertical levels. The sea surface temperatures and sea ice are prescribed with daily reanalysis data (Reynolds et al., 2007). The model simulations are nudged every 6 h by ERA-Interim horizontal wind fields above 1500 m (Telford et al., 2008), while the temperature is not nudged to allow the fast adjustments by the BBA, following the recommendations of Zhang et al., (2014). The kappa-Kohler activation scheme is used in the model, with the kappa value of organic carbon (OC) $\kappa_{org}$ set to 0.3 (Chang et al., 2010). To decompose the BBA effect into radiative and microphysical effects,
we performed six simulations from 2016 to 2017, one with present GFAS BBA emission as the baseline simulation, and one has the same setting but $\kappa_{org}$ is 0; two without BBA emission for $\kappa_{org}$ set to 0.3 and 0, and two with BBA emission for $\kappa_{org}$ set to 0.3 and 0 but with the BBA absorption turned off (through modification of the refractive indices). Then the radiative and



microphysical effects of BBA are separated as Lu et al., (2018). However, the way to isolate the BBA radiative effect in this paper is slightly different, as our model only allows us to switch off the absorption of BBA. This assumes the cloud adjustment due to BBA scattering is negligible in our experiments. Also note the microphysical effect of BBA decomposed from our setting is driven by the variation of $\kappa_{org}$, thus the small fraction (around 10%) of OC from no-biomass burning emission in this

region (figure S1) would contribute a small error. Then the BBA radiative effect is further decomposed into direct, indirect (effective radiative forcing from BBA–cloud interactions, defined as rapid adjustments and the net forcing with these adjustments from BBA-cloud interactions), and semi-direct effects by the method of Ghan et al., (2012) and Gordon et al., (2018). Two years are simulated in the model (2016 and 2017), the averages during July and August are used, for consistency with the flight campaigns, to represent the BBA effects during the African fire season.

To evaluate the model simulated cloud fields and BBA plume over SEA, the aerosol extinction from ORACLES (2016, 2017) and CLARIFY are used to compare with the model data. For ORACLES, we use the aerosol scattering and absorption coefficients from TSI nephelometers and Particle Soot Absorption Photometer (PSAP) (Pistone et al., 2019); The extinction coefficient measured from Cavity Ring-Down Spectroscopy (Langridge et al., 2011) is used for CLARIFY. For the comparison,

we collocate the three-hourly variables from the baseline model simulation with the aircraft observations (Watson-Parris et al., 2016, 2019). Two different collocation are performed, one to the 4-D coordinates of the observations (time, longitude, latitude, altitude), and another one with 3-D coordinates (time, longitude, latitude), to provide model profiles at the location of the observations. The aerosol optical depth (AOD) at 550 nm from the Moderate Resolution Imaging Spectroradiometer (MODIS) Terra (MOD08_D3, Version 4.4) and Aqua (MYD08_D3, Version 4.4) level 3, 1° × 1° resolution, collection 6 daily products

are also used to further evaluate the model performance.

## 3 Results

### 3.1 Model evaluation

The spatial and vertical distribution of the BBA plume is critical to the aerosol-cloud interactions, as it can significantly impact the sign and the magnitude of the BBA effects (Bellouin et al., 2019). To evaluate the performance of the model, the spatial

and vertical distribution of aerosol extinction coefficient from the model are compared with the aircraft observations

[insert figure 1 here]

The mean spatial distributions of the aerosol extinction coefficient along the flight tracks are illustrated in Fig 1. Note that the modelled extinction is for ambient aerosols, while the measurement gives dry extinction. Although this inter-comparison is widely used in model studies (Shinozuka et al., 2019), it is a potential source of error for model / measurement discrepancies,

as the extinction coefficient will generally be larger in the model. From figure 1, the model generally agrees well with the measurements, and it captures the extinction coefficient peak around 2° W; however, it also overestimates the extinction around



5°W. Extinction coefficients are slightly underestimated by the model near the coast of southern Africa and overestimated over the SEA. These errors suggest that the reproduced plume generally agrees well with measurements but is transported too far north and west. These biases might be partly attributable to the coarse model resolution and the use of 3-hourly output, which reduces reliability in the collocation. The comparison of mean September AOD of the model and retrievals (at ambient

relative humidity) from the MODIS satellite instrument further confirms this bias (Figure S2), which indicates that the model error may be related to the location and initial altitude of biomass burning emissions. Furthermore, the BBA deposition in the model may be biased low.

The mean vertical distribution of the aerosol extinction coefficient is shown in Fig 2. The model extinction coefficient profile

is collocated to the 3-D (latitude/longitude/time) coordinate of the observation. It can be seen in the figure that the plume is above clouds from the coast to 2° W, where it shows the extinction peak. From east to west, the plume gradually subsided and came into contact with the clouds. At 5° W, the plume is generally inside the clouds; thus, the BBA can interact and modulate the cloud properties. This finding is also confirmed by previous studies (Adebiyi et al., 2015; Chand et al., 2009; Deaconu et al., 2019; Gordon et al., 2018). From figure 2, the modelled vertical distribution of BBA plume agrees quite well with the

measurements, with the measured peak extinctions generally captured by the model. However, near 11° W, the modelled extinction coefficient has a slightly lower altitude than the measurement. This may indicate that the altitude of the plume is lower in the model, i.e., the model has less aerosol above cloud or aerosol reaches lower when in clear sky, or it may be the result of comparing simulated extinction at ambient humidity to observations of dry extinction.

[insert figure 2 here]

This comparison shows that the model has skill in reproducing the BBA plume, although the plume is transported slightly too far west and north, and also at a lower altitude towards the western part of the region of interest (westward of 5° W). The bias of the BBA plume location and vertical profile reproduced by the model will contribute to the uncertainty of the BBA microphysical effect over the ocean west of 5° W and of the BBA radiative effect. However, these errors are relatively small as the BBA plume is generally well-simulated in the model, allowing us to investigate the BBA effect on the underlying and

interacting cloud and the radiation balance.

## 3.2 Biomass burning aerosols impacts on clouds

BBA can serve as CCN and further impact the CDNC and cloud optical depth. Meanwhile, it also has a significant impact on the atmospheric thermal structure and therefore, the maximum cloud supersaturation (SS), LWP and cloud albedo. The BBA effects on clouds are decomposed as radiative (absorption and scattering) and microphysical effects (detailed in section 2), and

their impact on the clouds is examined in this section.

### 3.2.1 Biomass burning aerosols effects on CCN

[insert figure 3 here]





CCN from BBA mainly occurs over land and in the eastern part of the SEA. From east to west, a sharp gradient of BBA CCN$_{0.2\%}$ (CCN at 0.2% supersaturation) is evident in Fig3 (a), which may be due to the strong aerosol wet and dry removal mechanisms over the SEA, resulting in only BBA with a very small diameter being transported so far away from the continent. Due to the low hygroscopicity of BBA, the small-diameter particles (below 0.1 μm) are unable to activate. Furthermore, these

fine particles decrease the average hygroscopicity of internally mixed aerosols, thus can reduce the CCN concentration. The budget of CCN$_{0.2\%}$ attributed to BBA account for ~ 68% of total CCN$_{0.2\%}$ in the cloud box (grey box) and ~ 50% in the whole domain (Figure S3), indicating that BBA is the dominant source of CCN in the marine stratocumulus deck area.

The BBA CCN$_{0.2\%}$ profile along the latitude of Ascension Island (Fig. 3b) shows a distinct gradient. With near-source

concentrations of 1000 cm$^{-3}$, the BBA CCN$_{0.2\%}$ are transported westward above the clouds and gradually enter the cloud layer from the cloud top, accompanying the increase of the marine boundary layer height and cloud height. These BBA could impact the cloud droplet number concentration either by acting as CCN or by evaporation of droplets through shortwave absorption. Although only a small fraction of the BBA associated to CCN$_{0.2\%}$ is contacted with cloud, the in-cloud CCN$_{0.2\%}$ can still reach up to ~ 500 cm-3, indicating the significant role of BBA acting as CCN and the potential impact upon the cloud and radiation

balance through modulation of CDNC.

### 3.2.1 Biomass burning aerosols effects on cloud droplets

[insert figure 4 here]

The July and August averaged profile of BBA radiative and microphysical effects on maximum supersaturation (SS), as diagnosed by the activation scheme, from 2016 to 2017 are illustrated in figure 3. BBA slightly increase SS near the continent

and at low altitude over the SEA, while decrease SS at the higher altitude. The increased SS mainly results from the BBA absorption effect, as the SS profile is shifted to a lower altitude over the ocean. This SS altitude shift may be related to the change of the MBL height (Figure S4). When BBA accumulates above the inversion the absorbed shortwave radiation warms the air at the bottom of the inversion layer, strengthening the temperature inversion and decreasing the MBL height. This is also supported by a radiosonde research (Adebiyi et al., 2015), which also found a shoaling of the boundary layer when

absorbing aerosol was above. This effect is especially notable further away from the continent, where the MBL is also higher and sensitive to the temperature profile variations. Near the coast, BBA are generally above the underlying cloud deck; the absorption aerosols could strengthen the boundary layer inversion and thus decrease the dry air entrainment resulting in increased humidity and hence SS. The increased SS due to BBA absorption can be up to 53 % of the total SS, indicating the significant role of the BBA absorption on the cloud droplet formation. The BBA scattering has little impact on the SS, with

the mean effect around 0. The microphysical effect of BBA always exerts a negative impact on the SS, as expected from BBA acting as a condensation sink through hygroscopic growth or CCN activation and subsequent droplet growth. However, the decrease of the SS due to the BBA microphysical effect is comparatively small, indicating that the ability of BBA acting as CCN in our simulations is limited by its low hygroscopicity. In general, the BBA total effect on the SS shares a similar pattern





with the absorption effect. However, as the BBA radiative and microphysical effect counterbalance in the lower part of the cloud, the total BBA effect on SS is smaller near the continent and at the cloud base. The decrease of SS from the BBA total effect is still quite noticeable.

[insert figure 5 here]

Before the onset of collision coalescence CDNC is determined by both the CCN and SS, and the variation of CDNC due to BBA is shown in figure 5. As illustrated through the previous analysis, although the radiative properties of BBA are not directly related to the CCN number concentration, this could still alter the SS and hence impact the activation of CCN. The change in CDNC due to the absorption of BBA shows a corresponding response to the effect of BBA on SS; shifting to lower altitude over the ocean, which expressed as increasing at the cloud base and decreasing at the cloud top over the ocean compare to the

baseline simulation. Interestingly, the BBA absorption increases CDNC up to 102 cm$^{-3}$ near the continent, which is surprisingly high as the SS only increases 0.152% by the absorption. This may partly be because the increased cloud fraction near the continent caused by the stabilising effect of absorption results in the increase of total CDNC; or the critical supersaturation of ambient aerosols is around the cloud SS, thus a slight variation of the cloud supersaturation would activate large amount of CCN. Unlike the effect of BBA absorption, the increased CDNC due to the microphysical effect is more notable over the sea,

because only when the BBA are entrained and interact with the cloud, it can be activated as cloud droplets. The scattering effect only slightly increases CDNC when the MBL is deep enough to entrain BBA (Fig.3). However, similarly to the BBA scattering effect on SS, the increased CDNC due to scattering is negligible. In general, the substantial increase of CDNC by BBA can be attributed to the combined effect of absorption and microphysics, where the former mainly increases CDNC near the continent and at the lower altitude, and the latter increases CDNC above the ocean. BBA contribute up to 56% of total

CDNC (Figure S6), which is less than the fraction of CCN$_{0.2\%}$ it contributed. This may indicate the maximum supersaturation achieved in the clouds is lower than 0.2% and most transported BBA have a small diameter; thus, the actual activated BBA are less than could be expected. However, the BBA attributed CDNC is still more than half, which confirm the primary source for the cloud droplets is biomass burning in this region.

### 3.2.2 Biomass burning aerosols effects on cloud liquid water

25                                     [insert figure 6 here]

The simulated changes of LWP in figure 6 shows a distinct response to BBA over the SEA. Within the cloud box area, the BBA interaction can increase LWP by up to ~34% of the total (Figure S7), indicating the critical influence of BBA on the stratocumulus deck. Figure 6 shows that the BBA impacts the LWP mainly through its absorption effect. The increased LWP due to BBA absorption is more significant near the continent than in other areas, which may be because most BBA are above

cloud near the continent. This finding is consistent with the result of the large-eddy simulation by Johnson et al. (2004) that above cloud BBA can inhibit cloud-top entrainment and increase LWP. When BBA is transported further from the continent, the entrainment of BBA into the cloud layer reduces cloud droplets numbers through its absorption effect, which further reduces the increase of LWP, and results in a nearly zero or slightly negative effect on LWP. As a result of the different effects





of the absorption by BBA as well as its spatial distribution (more concentrated near the continent), the LWP from BBA absorption shows a steady negative gradient from west to east, indicating the role of BBA in modulating the cloud distribution. The microphysical effect of BBA, which is less clearly distinguishable, generally increases the LWP above the ocean. However, the increase of LWP by the BBA microphysical effect in the cloud box only accounts for ~ 4% of the total LWP, far less than the BBA absorption effect. Therefore, the BBA effect on the LWP is mainly due to its absorption characteristics.

### 3.2.3 Biomass burning aerosols effect on cloud albedo

[insert figure 7 here]

Cloud albedo is crucial in climate, as it is one of the critical parameters in determining the shortwave cloud radiative effect. As shown in figure 7, BBA generally increases cloud albedo in the cloud box area (total effect), which is consistent with relationships derived from a satellite based analysis (Deaconu et al., 2019). The cloud albedo increased by BBA account for ~8% of the total in the area where the stratocumulus cloud deck dominates (cloud box area) (Figure S8). The effect of BBA on cloud albedo from BBA can be primarily attributed to absorption and the microphysical effect; these two effects together can account for the ~90 % of the cloud albedo increase due to BBA in the cloud box area. Unlike the microphysical effect, BBA absorption significantly increases cloud albedo near the continent where most BBA are above the cloud. The above cloud BBA can decrease the dry air entrainment and increase the liquid water content due to absorption (cf. Fig. 6), and lead to an increase in cloud particles and higher cloud albedo. However, when more BBA are entrained into the MBL, the BBA decrease the number of cloud droplets and therefore have a negative impact on the cloud albedo. Therefore, the two different effects of BBA absorption – BBA above clouds and inside clouds – counteract each other and result in a slight increase of LWP and a near-zero impact on the cloud albedo near the western boundary of the cloud box. Note that the LWP and the cloud albedo changes are consistent, although the different colour scale and the non-linear response of cloud albedo to LWP may result in the cloud albedo having less variation than the LWP in the western boundary of the cloud box. The microphysical effect of BBA increases cloud albedo homogenously over the ocean, because the increase of CCN provided by BBA increases CDNC. Compared to the effect of BBA absorption, the increased cloud albedo due to a change in CCN is small, indicating again the significant role of the BBA radiative properties.

### 3.3 Biomass burning aerosols radiative effect

[insert figure 8 here]

The time-averaged BBA effects on the top-of-atmosphere radiation balance are investigated in this section. The simulated direct radiative effect of BBA generally is positive, except in the western areas of the ocean (northwest of Ascension Island), where the BBA have transported far away from its source. The different sign of the mean direct effect depends on the underlying surface brightness; thus, when BBA are above clouds, the direct effect shows a warming effect while, when at clear sky, far away from the continent, it shows a cooling effect. However, the cooling due to the direct effect is negligible, as only a minor proportion of BBA with small particle diameters are transported so far west. The July-August averaged warming effect





from the direct effect is large in the cloud box area: up to ~25.5 W m$^{-2}$ near the continent. The indirect radiative effect of BBA shows a similar pattern to the LWP changes due to the microphysical effect of BBA, and has a July-August mean cooling effect of -1.2 W m$^{-2}$ in the cloud box area. In some areas, the indirect effect shows a slight warming effect, which may be due to the variation of meteorological conditions such as free-tropospheric humidity and lower tropospheric stability, as these can

have prominent effects upon the magnitude and the sign of indirect effect (Ackerman et al., 2004; Chen et al., 2014). The magnitude of the indirect effect is strongly related to the CCN; particles with high hygroscopicity could further increase the CDNC. Thus, different settings of OC hygroscopicity would result in differences in the indirect effect. In this paper, we use a kappa value of 0.3 for OC, which may account for some of the uncertainty in the indirect effect.

The BBA semi-direct radiative effects show the most substantial cooling in the cloud box; however, they also have a warming effect in the northwest areas over the sea outside the cloud box. The July-August semi-direct effects can be up to ~ -52 W m$^{-2}$ near the coast, and dominate the total radiative effect in the cloud box area. The cooling of the semi-direct effects is mainly located in the area where the BBA are above the clouds and results from the significant increase of LWP and cloud albedo in that area (due to the stabilising effect of BBA absorption). The warming effect dominates where the cloud fraction is low, and

BBA have already entered the boundary layer, which further reduced the cloud fraction and leads to the positive semi-direct effects. Thus, as the dominant effect over the Southeast Atlantic, the magnitude and the sign of the semi-direct effects are strongly dependent on the relative location of the BBA and the cloud layer. Herbert et al., (2020) studied different layers of the plume with different altitudes, and find out the closer the aerosols layer to the cloud top, the stronger the magnitude of the semi-direct effects. However, in our simulation, the BBA plume is not well separated from the underlying clouds. Thus, when

the absorption aerosols closer to the cloud, some BBA may have entered the cloud layer. The semi-direct effects is resulted from both above cloud cooling and below cloud warming.

The total net radiative effect of the BBA shows a similar spatial pattern to the semi-direct effects albeit with a smaller magnitude, reflecting the dominant role of the semi-direct effects in this region. The total July-August BBA radiative effect

over the whole domain is -0.9 Wm$^{-2}$, exerting a net cooling effect in that area. In the cloud box, the July-August averaged BBA total radiative effect can up to -30 Wm$^{-2}$, with a mean value of -5.7 Wm$^{-2}$. Gordon et al. (2018) have previously estimated the BBA radiative effects near Ascension Island using the same model with a different high-resolution configuration and model version. The direct and semi-direct effects show good agreement between our simulations and their findings; however, their results (direct effect: 10.3 Wm$^{-2}$ and semi-direct effects: -16.1 Wm$^{-2}$) are slightly higher than our simulated cloud box mean

values, as they only sampled the five most polluted days during their simulations. Nevertheless, the indirect effect in their results is -11.4 Wm$^{-2}$, which is much higher than our simulation. The possible reason behind this discrepancy is that the OC kappa value in their simulation is 0.88, which is much higher than our setting of 0.3. Furthermore, the meteorological conditions are different as they only averaged five days.

[insert figure 9 here]

The mean BBA radiative effects in the shortwave and longwave are summarised in the figure. 9. In the cloud box, the semi-direct effects are the dominate BBA radiative effect, resulting in a considerable cooling of the total radiative effect over the cloud area. The cooling of semi-direct effects in the cloud box is generally at the shortwave, while at longwave, semi-direct effects show a slight warming effect. This may result from the semi-direct effects enhancement of LWP and cloud cover,

which would further increase the sunlight reflection as well as the absorption of longwave radiation from the underlying warmer surface. The direct effect is 7 Wm$^{-2}$ in the cloud box area, which partially cancels the cooling of the semi-direct effects. The indirect effect is cooling in this area. However, its magnitude is relatively small, which may results from the limited capability of BBA in acting as CCN due to its low hygroscopicity.

For the regional domain, the BBA semi-direct effects also show a negative cooling effect. However, compared with the cloud box, the mean value of semi-direct effects decreases rapidly when the averaged domain size increases, as it is only about -1.6 W m$^{-2}$ for the regional domain, i.e. ~ 13% of the semi-direct net effects in the cloud box area. Globally, the net semi-direct effects are nearly zero, indicating the semi-direct effects from biomass burning primarily affect the cloud deck over the SEA. The regional averaged indirect effect is similar to the cloud box mean, and slightly lower than the regional semi-direct effects,

indicating the role of the BBA cloud interactions in this region. In general, BBA have the most significant radiative effects in the cloud deck area, followed by in the South Atlantic Ocean and west African (regional domain). The indirect effect is generally the same in these areas and is one of the critical factors in determining the regional radiation balance. The dominant effect in these areas is the cooling effect exerted by the semi-direct radiative effects.

### Discussion and conclusion

The UK Earth System Model (UKESM1) is used to investigate the effects of biomass burning aerosols over the southeast Atlantic to provide both a better understanding of their radiative and microphysical effects on clouds, and the radiation balance in this area. The analysis focuses on the biomass burning seasons from July to August for the years 2016 and 2017, which facilitates model evaluation with flight measurements from the ORACLES and CLARIFY measurement campaigns.

Comparison with the flight observations shows that the model generally captures the spatial and vertical distributions of BBA plume; however, the simulated plume is located too far north-west and at a slightly lower altitude in the model. Although the semi-direct effects and cloud response are sensitive to the relative distance of cloud and biomass burning plume (Herbert et al., 2020), these errors are relatively small, providing the foundation for our investigation of the BBA effect on clouds and the radiation balance in this region.


The BBA associated CCN are emitted from the land and then transported westward above the cloud. With the increase of the marine boundary layer height, and reduction of the plume height, BBA enter the cloud layer from the top. The budget of



$CCN_{0.2\%}$ attributable to BBA can account for $\sim 68\%$ of the total $CCN_{0.2\%}$ in the cloud box area, indicating that BBA are the primary source of CCN for the marine stratocumulus deck.

The effects of BBA on clouds are separated into radiative effects (including the effects from absorption and scattering) and the microphysical effect. The impact of BBA on in-cloud maximum supersaturation is mainly due to its absorption. When BBA accumulate above the inversion, the absorbed shortwave radiation warms the air at the bottom of the inversion layer, lowering the temperature inversion and decreasing the marine boundary layer height. As a consequence, the maximum supersaturation shifts to a lower altitude above the ocean. Near the coast, the above cloud BBA strengthens the temperature inversion, which results in the weakening of the entrainment across the inversion layer, as buoyant parcels of air in the MBL require more energy in order to push through the strengthened temperature inversion (Herbert et al., 2020). Therefore, the relative humidity increases, as well as the supersaturation. The microphysical effect decreases maximum supersaturation as BBA can act as CCN and allow additional water vapour to condense; however, this decrease is comparatively small in this area. Due to the shift of maximum supersaturation by BBA absorption the CDNC shows a corresponding response: increasing at low altitudes (cloud bottom in baseline simulations) and decreasing at high altitudes (cloud top from baseline) over the ocean. However, the BBA microphysical effect increases CDNC further from the continent, cancelling out the decreases from absorption. The CDNC attributed to BBA can be up to 56% of total CDNC, confirming the significant impact of BBA on the cloud deck.

The BBA absorption effect increases LWP significantly when BBA are located above the stratocumulus deck, as the stabilisation from absorption can inhibit cloud-top entrainment. When BBA enter the cloud layer, it can decrease the amount of condensable liquid water and so decrease the LWP. As a result, the variation of LWP due to the absorption effect is nearly zero or slightly negative when far away from the continent. The microphysical effect also contributes to the increase in LWP; however, this increase is small compare to the absorption effect. Therefore, the LWP response to BBA is dominated by the effect of absorption, showing a substantial increase over the Southeast Atlantic. The variation of cloud albedo due to BBA shows a similar pattern to the LWP.

The dominance of the effect of absorption on cloud properties is reflected in the effect on the top-of-atmosphere radiation balance. When the BBA are above the stratocumulus deck, semi-direct effects contribute most to the overall cooling, while they also exert a warming effect in the northwest areas over the sea. The magnitude and the sign of the semi-direct effects are dependent on the relative location of BBA and clouds, as BBA can either increase the underlying cloud LWP or decrease the surrounding droplet numbers depending on whether the BBA are above or inside the cloud. The direct radiative effect is generally positive and shows a strong warming when BBA are above the stratocumulus deck (with July-August average 7.5 W m$^{-2}$), as the surface albedo of the underlying clouds is fairly high. However, for the total net BBA radiative effect the positive direct radiative effect is more than compensated by the semi-direct effects, resulting in an overall cooling effect over the SEA (with July-August average -0.9 W m$^{-2}$). In addition to the semi-direct effects, the indirect radiative effect is also negative,



showing a cooling in this area. The indirect effect mainly results from the response of LWP to the BBA microphysical effect, as they share a similar spatial pattern. When comparing the BBA radiative effects at different scales, we find that semi-direct effects from biomass burning play a significant role over the southeast Atlantic stratocumulus deck, while it has little impact in the global mean. The indirect effect from biomass burning aerosol, however, have a similar magnitude in both regional and

global, showing a more widespread cooling effect.

**Data availability**

The original simulation data are available from JASMIN facility upon request. There is also processed model data, which can be downloaded from https://data.mendeley.com/datasets/xdxh8stc48/2. Data from the CLARIFY aircraft campaign are available on the CEDA repository http://archive.ceda.ac.uk/. Data from ORACLES aircraft campaigns are available on the

repository https://espo.nasa.gov/oracles/archive/browse/oracles.

**Author contributions**

PS and HC developed the concepts and ideas for the direction of the paper. HC and HG set up the model. HC carried out and analysed the model simulation. DWP and HC performed the model validation, LD, DWP, HG, HC and PS contributed the analysis of the results. HC wrote the paper with input and comments from all other authors.

**Competing interests**

The authors declare that they have no conflict of interest.

**Special issue statement.**

This article is part of the special issue "New observations and related modelling studies of the aerosol- cloud-climate system in the Southeast Atlantic and southern Africa regions (ACP/AMT inter-journal SI)". It is not associated with a conference.

**Acknowledgements**

This research has been funded by the NERC CLARIFY project NE/L013479/1. P.S. additionally acknowledges support from the European Research Council (ERC) project constRaining the EffeCts of Aerosols on Precipitation (RECAP) under the European Union's Horizon 2020 research and innovation program with grant agreement 724602. We sincerely acknowledge Kate Szpek from Met Office CLARIFY team, and Steven Howell from NASA ORACLES team for providing aerosols
scattering and absorption data, and all CLARIFY and ORACLES science teams for data support. We also thanks Luke



Abraham and William Ingram for the help with the model, Ben Johnson for the help of setting up biomass burning emissions, William Jones for proofreading the manuscript. We acknowledge the use of the Monsoon2 system, a collaborative facility supplied under the Joint Weather and Climate Research Programme, a strategic partnership between the UK Met Office and the Natural Environment Research Council (NERC). We also used the JASMIN facility (http://www.jasmin.ac.uk/) via the Centre for Environmental Data Analysis, funded by NERC and the UK Space Agency and delivered by the Science and Technology Facilities Council.

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





**Figures**

.

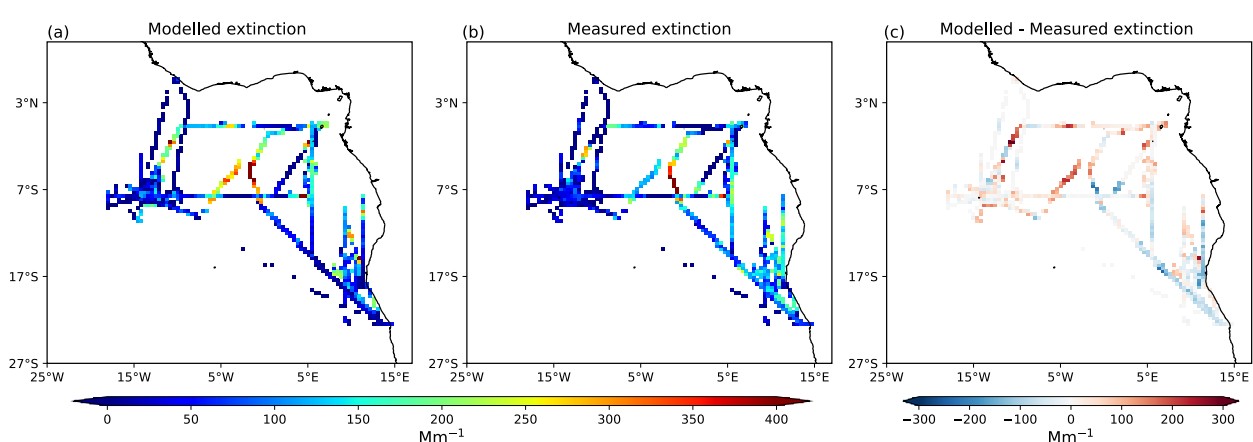

Figure 1: Mean (a) modelled and (b) measured aerosol extinction coefficient [Mm$^{-1}$] along the flight tracks and the (c) differences between model and measurement. Note that the model extinction is at ambient conditions whereas the measured extinction is for dry aerosols with relative humidity below 30%.

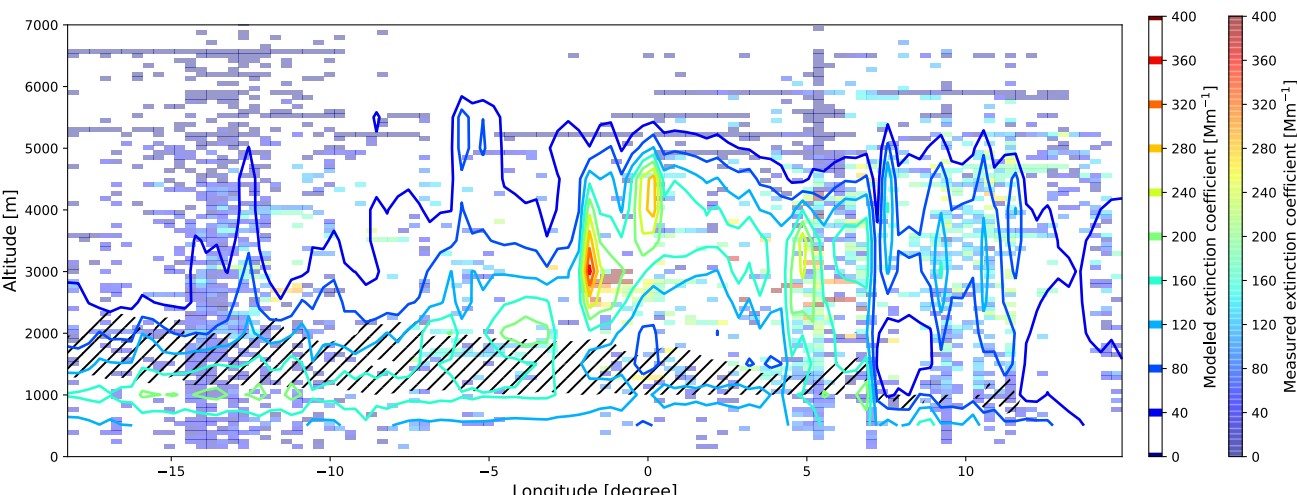

5 Figure 2. Mean along-flight track vertical distribution of the aerosol extinction coefficient along longitude. The contour lines
show the mean collocated model extinction coefficient profile along with the location of the aircraft. The pixels represent the
mean value of aerosol extinction coefficient from CLARIFY and ORACLES (2016, 2017) campaigns. The hashed lines
illustrate the model cloud location by using cloud liquid water content from the model. Note that the modelled extinction is
for ambient relative humidity whereas the measured extinction is for dry aerosols with relative humidity below 30%. The same
10 colourmap is applied for measurement and model result to facilitate comparison.

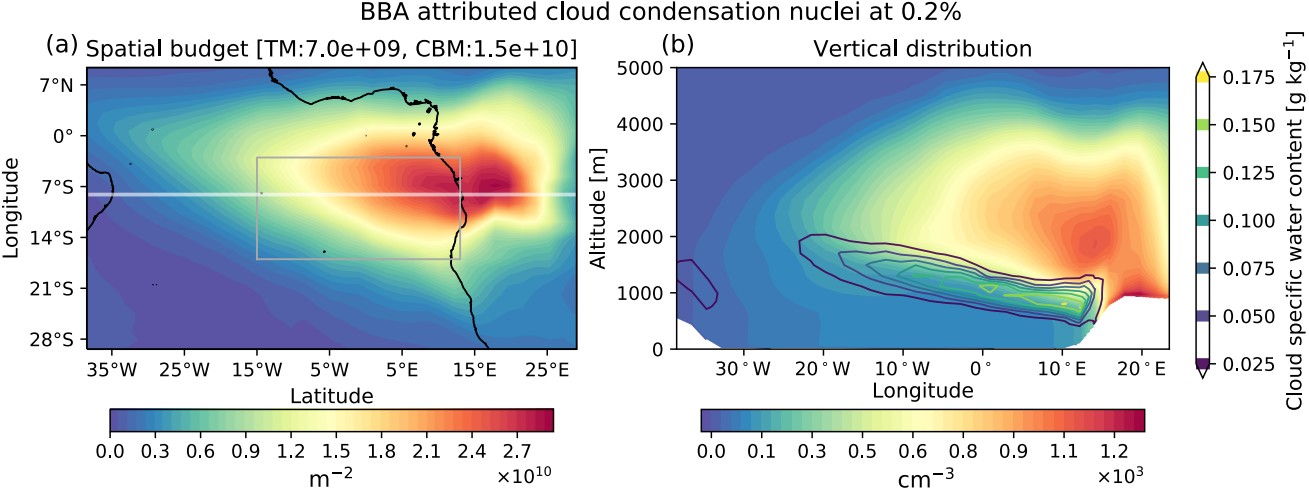

5  Figure 3. UKESM1 simulated mean cloud condensation nuclei attributed to BBA at 0.2% supersaturation under standard

conditions for temperature and pressure (STP) during July and August 2016-2017 as (a) the vertically integrated burden and

(b) profile along the latitude of Ascension Island, 8.1° S (the white line in Fig. 3a). The domain in Fig. 3a, ranging from 30° S

to 10° N and from 40° W to 30° E, is the areas this paper interested in. The grey box in the map (cloud box) representing the

cloud areas where the averaged low cloud fraction is above 0.58. The TM is the total mean of the domain and the CBM is the

10  mean of the cloud box. The contours in Fig. 3b are the cloud specific water content in the baseline simulation.



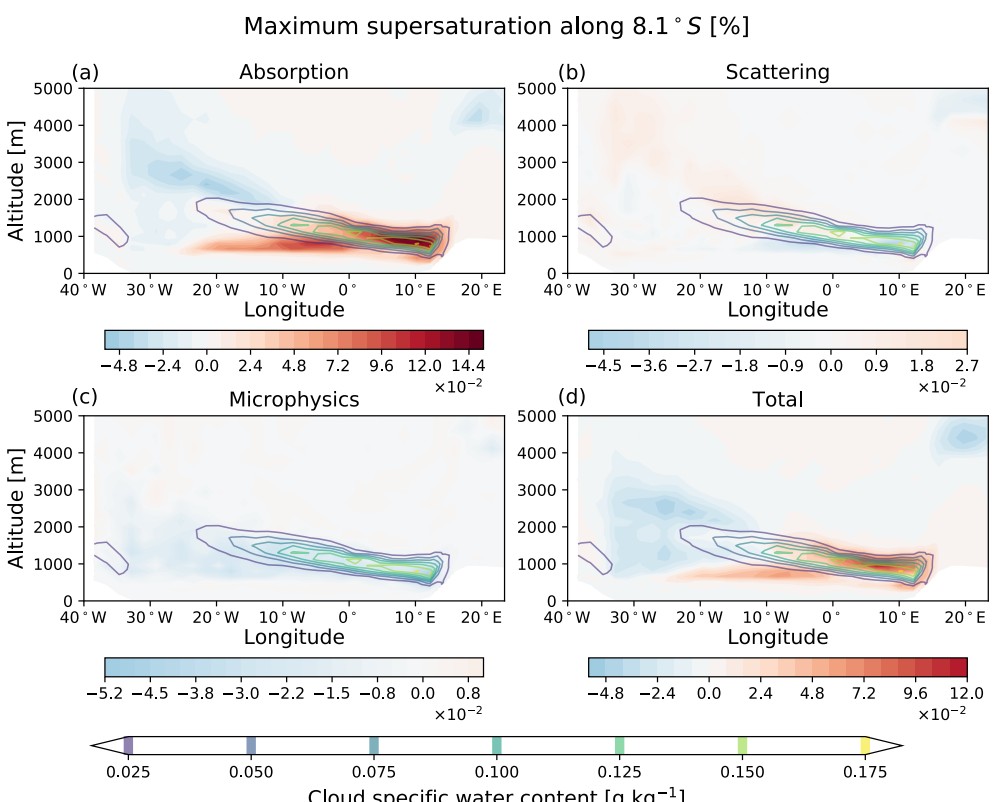

Figure 4. UKESM1 simulated mean vertical profiles of the BBA effects (a) absorption, (b) scattering, (c) microphysical and (d) total on maximum supersaturation along the latitude of Ascension Island (cf. Fig. 3a) during July and August, 2016-2017. The contour lines are the baseline cloud specific water content. The same colourmap scale is used in each plot to facilitate comparison, but the colourmap ranges differ in each plot, corresponding to the maximum and minimum of SS at each.





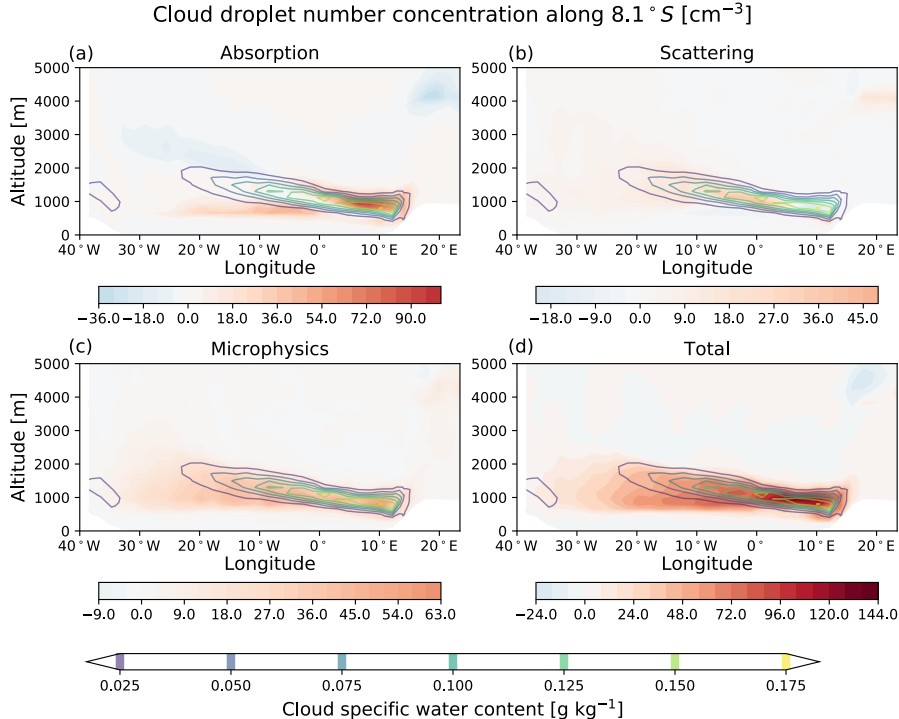

Figure 5. Same as Figure 4 but for the in-cloud cloud droplet number concentration per cubic centimetre.

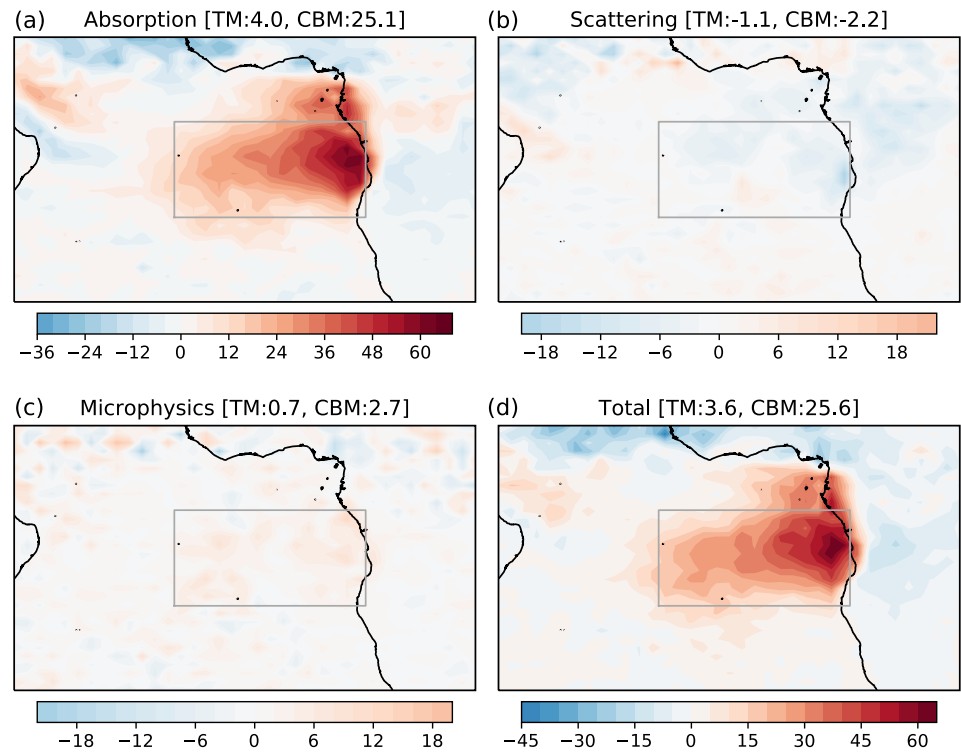

Figure 6. UKESM1 simulated mean spatial distribution of the BBA effects of (a) absorption, (b) scattering, (c) microphysical and (d) total on the cloud liquid water path during July and August, 2016-2017. The domain range is from 30° S to 10° N, and from 40° W to 30° E. The TM is the total mean of the domain and the CBM is the mean of the cloud box (the grey box on the map) representing the areas where the average low cloud fraction is above 0.58. The same colour scale is used in each plot to facilitate comparison, but the colourmap ranges differ in each plot, corresponding to the maximum and minimum variation of LWP in each.



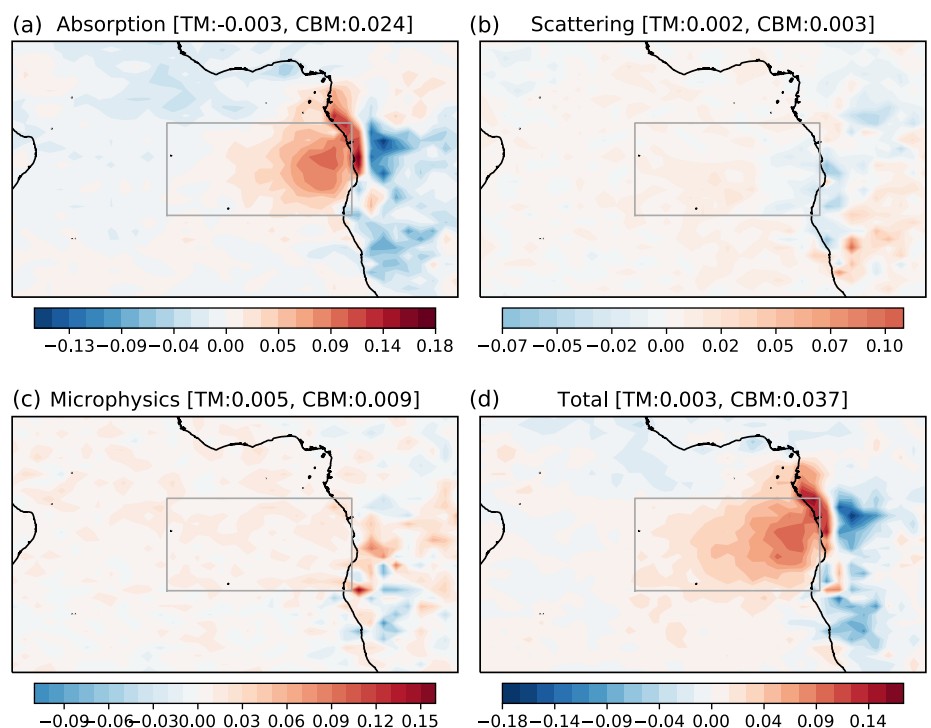

Figure 7. UKESM1 International Satellite Cloud Climatology Project (ISCCP) simulator mean spatial distribution of the BBA effects of (a) absorption, (b) scattering, (c) microphysical and (d) total on the cloud albedo during July and August, 2016-2017. The domain range is from 30° S to 10° N, and from 40° W to 30° E. The TM is the total mean of the domain and the CBM is the mean of the cloud box (the grey box on the map) representing the areas where the average low cloud fraction is above 0.58. The same colour scale is used in each plot to facilitate comparison, but the colourmap ranges differ in each plot, corresponding to the maximum and minimum variation of cloud albedo in each.





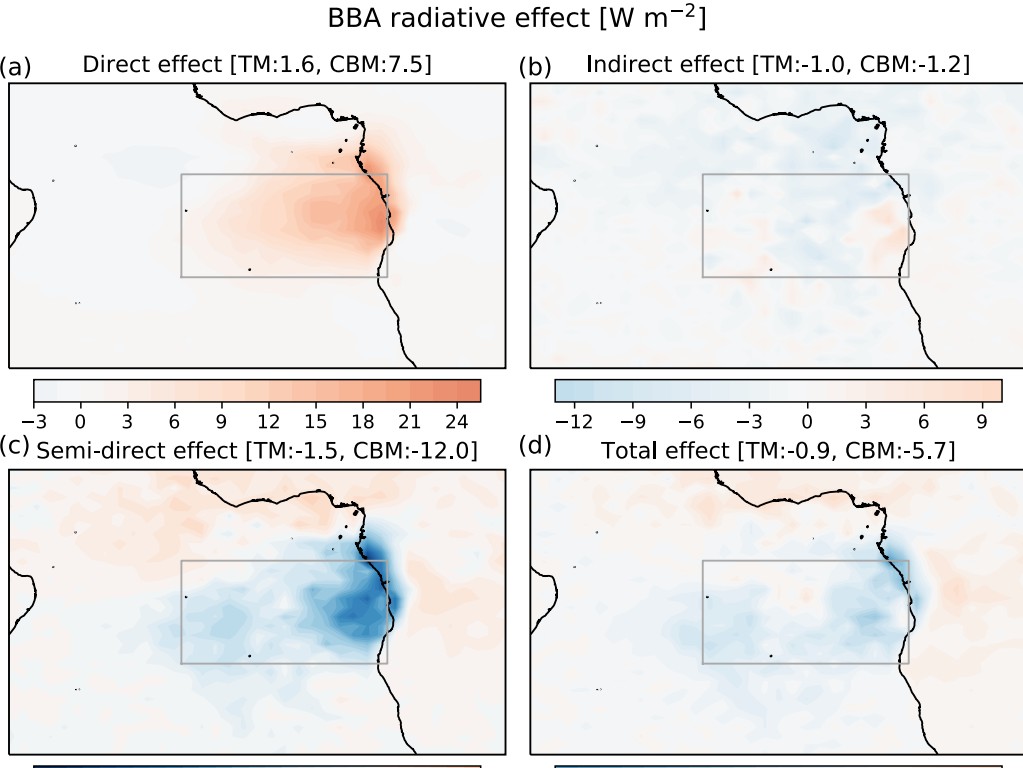

Figure 8. UKESM1 mean net (shortwave + longwave) biomass burning aerosols (a) Direct, (b) indirect, (c) semi-direct, and (d) total radiative effects during July and August, 2016-2017. The same colourmap scale is used for each plot, but the colourmap ranges differ in each plot, corresponding to the maximum and minimum of the effect in each.



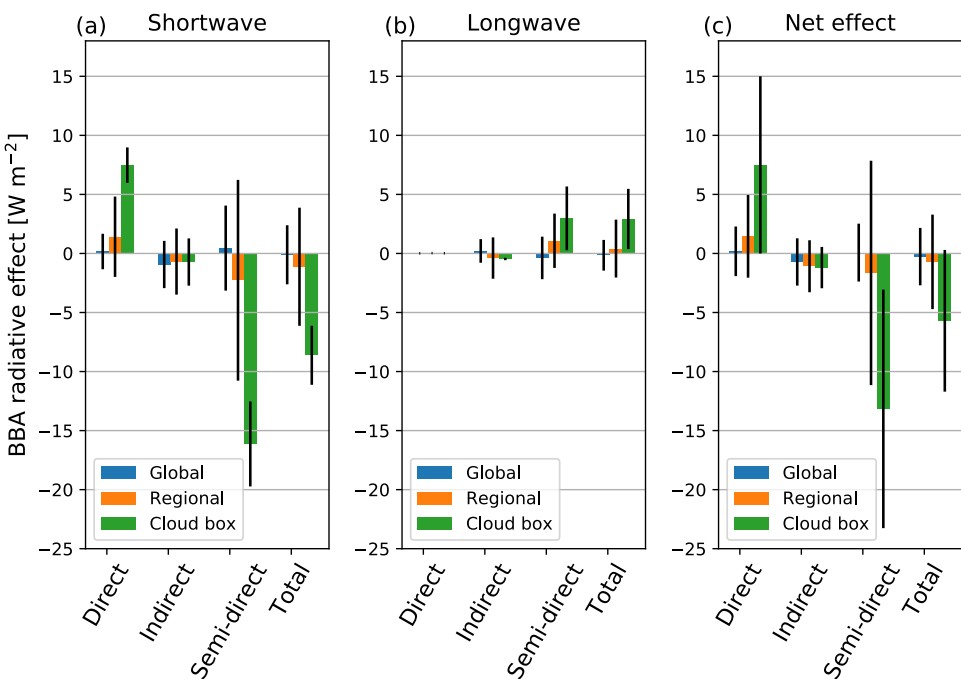

Figure 9. Bar chart of UKESM1 mean BBA radiative effect during July and August, 2016 to 2017. The BBA radiative effect at (a) shortwave, (b) longwave, and the (c) net effect are presented in separate plots. The blue colour represents the global mean, the orange is the domain mean, and the green is the cloud box mean. The error bars represent standard errors.

