# Peer review of "Cloud adjustments dominate the overall negative aerosol radiative effects of biomass burning aerosols in UKESM1 climate model simulations over the south-eastern Atlantic"

_Atmospheric Chemistry and Physics, 2020_

## Referee Comment (RC1) · Michael Diamond (Referee) · 29 Jul 2020

In this manuscript, Che et al. run the latest generation of UKESM to investigate aerosol direct, semi-direct, and indirect effects from biomass burning smoke plumes produced by agricultural burning over the southeast Atlantic Ocean. The headline finding (in my read) is that the semi-direct effect of cooling due to increased cloudiness from a stronger cloud-top inversion dominates the overall radiative forcing, offset substantially by the direct effect of smoke absorption and reinforced marginally by indirect effects.

The manuscript is well organized and the findings appear sound for the most part. (I do have some questions below, mainly pertaining to the Twomey effect). The text

could use some areas of clarification and potentially additional information. I would not anticipate any major new analyses would need to be undertaken to address my comments, so I therefore recommend publication following minor revisions.

General comment:

My biggest (really, only) concern with the results is that the indirect effect estimate seems unrealistically small given the change in cloud droplet number concentration you diagnose. From a simple back-of-the-envelope calculation of the Twomey effect, a doubling of cloud droplet concentration (you report 56% of CDNC are from biomass burning) should lead to a radiative forcing of O(10) W/m2, as in Lu et al. (2018). This is before taking into account the small liquid water increases you find. In my own work in the region (Diamond et al., 2020), I've estimated that a 5% increase in climatological CDNC from the influence of shipping produces a radiative forcing of $\sim$-2 W/m2 during austral spring. I thus find it hard to believe that doubling CDNC would produce less than that in austral winter.

Specific comments:

1. Title: The title is currently rather non-informative. I would recommend highlighting that the semi-direct effects dominate (really the highlight of the paper in my opinion) in the title, and also mentioning that this is a global climate modeling study (as opposed to in situ or satellite observations or high-resolution process modeling).

2. Page 2, Line 1: I don't follow why the net sign of the aerosol radiative effect, rather than its magnitude, signifies the importance of aerosol in this region.

3. Page 3, Line 3: I would argue that in situ results showing abundant biomass burning influence within the MBL at Ascension Island from the LASIC campaign (Zuidema et al., 2017) and throughout the SE Atlantic Klein-Hartmann box over the remote ocean from ORACLES (Diamond et al., 2018; Kacarab et al., 2020) are more directly relevant to your point about smoke-cloud interaction.

4. Page 3, Line 5: I don't believe the cited literature backs up the claim that "most" of the BBA is entrained into the MBL.

5. Page 3, Line 24: The large eddy simulation results of Yamaguchi et al. (2015) and Zhou et al. (2017) also seem relevant to cite/discuss here.

6. Page 4, Lines 1-2: The BBA plume subsiding and gradually meeting the rising MBL is true in the mean, but the picture is much more nuanced in reality, as instances of smoke-cloud contact were seen to be highly variable between and even with flights during the ORACLES and CLARIFY campaigns. It is also not necessarily the case that smoke-cloud contact corresponds instantaneously with the MBL being polluted, as discussed in Diamond et al. (2018).

7. Page 4, Line 13: Is dust included as one of the "five interactive log-normal aerosol modes"? I only count four other components (sulfate, sea salt, black carbon, organic carbon). The phrasing currently is confusing, as it sounds like the dust representation is entirely separate.

8. Page 5, Line 9: Somewhere in the manuscript you should discuss the implications of only looking at one part of the biomass burning season (July-August). It is well known that the BB plume properties change over the course of the biomass burning season, influenced in part by meteorological shifts like the strengthening of the southern African Easterly Jet (AEJ-S) in September and October that corresponds with a more elevated plume (Adebiyi & Zuidema, 2016).

9. Page 5, Lines 17-20: I'm surprised that you do not use any of the new products from MODIS or SEVIRI that account for above-cloud aerosol absorption. I would recommend trying the comparison using one of those products or at least discussing the issues with traditional AOD products that cannot retrieve AOD in the presence of clouds.

10. Figure 1: What altitude is being shown, or is this a column average? There is a

large amount of vertical variability in the plume (as seen in Figure 2) so the 2D picture is a bit difficult to interpret.

11. Page 6, Line 4: I do not understand why you only compare September AOD when the analysis focuses on July-August. As discussed earlier, there are known differences in plume location throughout the biomass burning season, in part driven by different meteorological factors between July-August and September-October. Thus, it's entirely possible that the model could represent one part of the season well but the other poorly if it is not representing those meteorological shifts properly. Figure S2 should be replaced with a new version including July and August.

12. Page 6, Lines 12-13: As discussed above, although this description makes sense in the climatology, the picture we found in the field is much more complicated than the mean suggests. For one, much of the smoke in the marine boundary layer at a given location may have been entrained upstream and not necessarily reflect the properties of the plume above-cloud at the time of sampling (Diamond et al., 2018). It may be worth noting that although the mean field shows a plume subsiding from east to west, actual plume distribution and occurrence of plume-cloud contact at any given time is more nuanced.

13. Page 7, Lines 6-7: Are these percentages for the column burden? It may be worth also reporting the values for the marine boundary layer separately (if they differ), as the MBL CCN concentration is what matters most for cloud droplet activation.

14. Figure S3: Figure S3 is an exact copy of Figure 3. I believe the figure the authors meant to include would show the change in CCN due to BBA?

15. Page 7, Line 27: This should be testable by looking at the average strength of the cloud-top inversion between the different model runs directly.

16. Page 8, Line 4: This statement needs qualification, as the SS increases where most of the cloud mass is. Are you only referring to the westernmost region? Or this

actually supposed to say that the increase in SS is noticeable in the net (decrease from microphysics is more than compensated by increase from absorption)?

17. Page 8, Line 20: BBA being 56% of the CDNC is less than the 68% figure quoted above for CCN, but is that for the column or MBL only? It would be more relevant to compare the fraction of MBL CCN that is from BBA to the CDNC change, as the BBA aloft does not activate.

18. Page 8, Line 30: The various LES studies cited in this review (Yamaguchi et al., 2015; Zhou et al., 2017) and by the authors (Herbert et al., 2020) seem relevant to reference here in addition to the classic study of Johnson et al. (2004).

19. Page 8, Line 33: This is due to the absorption effect lowering the relative humidity within the MBL, correct? It would be helpful to be explicit about this.

20. Page 9, Lines 2-3: The LWP effect of BBA absorption is to increase LWP as one moves from west to east. The text is written to make it sound as if LWP is decreasing from west to east due to BBA absorption. The text should be clarified here.

21. Page 9, Lines 16-17: You should clarify that the BBA in the MBL suppresses CDNC through the semi-direct effect here, not the indirect effect (which actually causes CDNC to increase substantially).

22. Page 9, Lines 23-24: As mentioned in the general comment, this result is very surprising given the large increase in CDNC, which should lead to an albedo increase of ∼0.05-0.10.

23. Page 10, Line 3: I don't understand how the indirect effects could have led to a warming given both CDNC and LWP increase. Did cloud fraction decrease anywhere? Or could this just be due to weather noise between different initializations?

24. Page 10, Lines 20-21: This sentence should be rewritten for clarity. The semi-direct effect is not cooling at cloud top and warming below; rather, above-cloud semi-direct effects lead to a TOA cooling whereas below-cloud semi-direct effects lead to a TOA

warming.

25. Page 10, Line 29: The results of Gordon et al. (2018) are also averaged over a different region that you are using, correct? It would be helpful to compare the values averaged over the same region, as the spatial mismatch could also lead to discrepancies.

26. Page 10, Lines 31-33: I would be more believing of this argument (the kappa values in Gordon et al., 2018, do seem unreasonably high) if you did not find a significant increase in CDNC even with your lower (and probably more realistic) kappa values in this study.

27. Page 11, Lines 4-6: If you're talking about TOA radiation, isn't the relevant effect that less OLR makes it out due to the radiation coming from the relatively cool cloud tops rather than the warmer surface? Zhou et al. (2017) discuss the potentially important role of LW radiative effects in BBA-cloud interactions.

28. Page 12, Lines 6-7. Increasing the inversion strength, rather than "lowering the temperature inversion"? Or are you talking about lowering the height of the inversion? I'd argue that has more to do with the clouds not being able to grow via entrainment

29. Page 12, Line 12: Cloud top/base is maybe not the most useful shorthand here, as the increase in SS occurs throughout the cloudy layer near the continent, where the cloud deck is most prevalent in general. The base/top difference only shows up further offshore.

30. Page 13, Lines 4-5: The global and regional indirect effects are "similar" in that they're both indistinguishable from zero... maybe you can argue the global effect is from long range transport and MBL advection, but I wouldn't necessarily highlight this idea in the very last sentence of your paper.

References:

Adebiyi, A. A., & Zuidema, P. (2016). The role of the southern African easterly jet in

Interactive
comment

modifying the southeast Atlantic aerosol and cloud environments. Quarterly Journal of the Royal Meteorological Society, 142, 1574-1589. doi:10.1002/qj.2765

Diamond, M. S., Director, H. M., Eastman, R., Possner, A., & Wood, R. (2020). Substantial cloud brightening from shipping in subtropical low clouds. AGU Advances, 1(1), e2019AV000111. doi:10.1029/2019av000111

Diamond, M. S., Dobracki, A., Freitag, S., Small Griswold, J. D., Heikkila, A., Howell, S. G., . . . Wood, R. (2018). Time-dependent entrainment of smoke presents an observational challenge for assessing aerosol–cloud interactions over the southeast Atlantic Ocean. Atmospheric Chemistry and Physics, 18(19), 14623-14636. doi:10.5194/acp-18-14623-2018

Herbert, R. J., Bellouin, N., Highwood, E. J., & Hill, A. A. (2020). Diurnal cycle of the semi-direct effect from a persistent absorbing aerosol layer over marine stratocumulus in large-eddy simulations. Atmospheric Chemistry and Physics, 20(3), 1317-1340. doi:10.5194/acp-20-1317-2020

Johnson, B. T., Shine, K. P., & Forster, P. M. (2004). The semi-direct aerosol effect: Impact of absorbing aerosols on marine stratocumulus. Quarterly Journal of the Royal Meteorological Society, 130(599), 1407-1422. doi:10.1256/qj.03.61

Kacarab, M., Thornhill, K. L., Dobracki, A., Howell, S. G., O'Brien, J. R., Freitag, S., . . . Nenes, A. (2020). Biomass burning aerosol as a modulator of the droplet number in the southeast Atlantic region. Atmospheric Chemistry and Physics, 20(5), 3029-3040. doi:10.5194/acp-20-3029-2020

Lu, Z., Liu, X., Zhang, Z., Zhao, C., Meyer, K., Rajapakshe, C., . . . Penner, J. E. (2018). Biomass smoke from southern Africa can significantly enhance the brightness of stratocumulus over the southeastern Atlantic Ocean. Proceedings of the National Academy of Sciences, 115(12), 2924-2929. doi:10.1073/pnas.1713703115

Yamaguchi, T., Feingold, G., Kazil, J., & McComiskey, A. (2015). Stratocumulus to

cumulus transition in the presence of elevated smoke layers. Geophysical Research Letters, 42(23), 10478-10485. doi:10.1002/2015gl066544

Zhou, X., Ackerman, A. S., Fridlind, A. M., Wood, R., & Kollias, P. (2017). Impacts of solar-absorbing aerosol layers on the transition of stratocumulus to trade cumulus clouds. Atmospheric Chemistry and Physics, 17(20), 12725-12742. doi:10.5194/acp-17-12725-2017

Zuidema, P., Sedlacek III, A. J., Flynn, C., Springston, S., Delgadillo, R., Zhang, J., . . . Muradyan, P. (2018). The Ascension Island boundary layer in the remote southeast Atlantic is often smoky. Geophysical Research Letters, 45, 4456–4465. doi:10.1002/2017gl076926
* * *

---

## Referee Comment (RC2) · Anonymous Referee #2 · 24 Aug 2020

This paper uses UKESM1 simulations to study the contribution of different processes to the radiative effects of biomass burning aerosols. The topic is important, the presentation quality is good and the paper shows interesting results. I don't have major concerns about the analysis, but I find there is the need for a more detailed description of the methodology and some additional analysis. I also believe that most of the supplementary figures belong to the main body of the paper (see specific comment below). The paper is worth publishing, but given that it needs additional work in a number of areas, I am recommending a major revision.

GENERAL COMMENTS

Only two years of model simulations are used. They are chosen because they coincide with observational campaigns. However, the observational data are only used to perform an initial assessment of the model's simulations and to justify the use of UKESM1 for the subsequent analysis, which is entirely model-based. Then, why not use a longer simulation period? This will allow to get more robust estimates of the BBA effects in the region, and to quantify the role of interannual variability.

The methodological description needs additional work. Especially, I think a brief description of how the different experiments are combined to decompose the BBA effect into individual contributions is lacking. How accurate is this decomposition? What are the caveats?

Given that this is a model-based study, the accuracy of the results will not only depend on the representation of the BBA plume and the cloud climatology, but also on how good the model is at representing the cloud response to the drivers of changes. For example, the realism of the strong radiative cooling of the semi-direct effect will depend on how well UKESM1 represents the cloud response to a strengthening of the inversion. Figure S4 touches on this, but only in passing. You cite a reference (Adebiyi et al., 2015) that uses radiosondes to look into this. How does the change in the inversion strength in UKESM1 compare to the one observed with radiosondes? I acknowledge that a comprehensive assessment of how well UKESM1 performs in this respect is out of the scope of this paper, but putting the UKESM1 changes into context would be very helpful.

The title should be more especific, and should capture the main message of the paper.

SPECIFIC COMMENTS

P3L22-P4L5: Most of this paragraph probably belongs to the methods section. Only the last sentence describes the objectives of the paper. I'd suggest transferring the description of the campaigns to the methods, and expand on what the paper is about.

[Figure]

Figure 1. The spatial resolution of the model results seem to be much higher than N96. Am I misinterpreting what that figure shows?

Figure 3 caption. "The domain in Fig. 3a, ranging from 30° S to 10° N and from 40° W to 30° E, is the areas this paper interested in." I believe the description of the area of interest belongs to the main text.

Figure 3 caption. " The grey box in the map (cloud box) representing the cloud areas where the averaged low cloud fraction is above 0.58." That would be the 0.58 isoline, unlikely to have a rectangular shape. Please provide a clearer explanation of what this box is.

P7L18. supersaturation (SS). This is an unfortunate acronym. In general, I don't see the need for using an acronym to compress a single word. For instance, we don't normally use MP for microphysics. Or, what would you use for subsaturation?

P9L2. "LWP from BBA absorption shows a steady negative gradient from west to east". I might be looking at the wrong region, but the gradient in the cloud region looks positive to me. Please clarify,

Figure 7. This figure shows changes in cloud albedo using the ISCCP simulator. I might have missed it, but I believe that the use of the ISCCP simulator is not documented in the methodology. Which simulator variables are used? Why is this approach better than looking at changes in cloud fraction and cloud radiative effect?

Discussion of Figure 8c. It would be nice to show a plot with the actual change in the strength of the inversion that drives this strong semi-direct effect.

Figures S5 to S8. I feel that these figures belong to the main paper, not to the supplementary material. They can be arranged all together in a multi-panel figure with showing the baseline climatologies, complementing the figures showing the changes due to different processes.

---

## Author Comment (AC1) · 19 Oct 2020

**Review replies to "The significant role of biomass burning aerosols in clouds and radiation in the South-eastern Atlantic Ocean" by Haochi Che et al.**

We would like to thank all reviewers for their constructive comments and suggestions on the manuscript. The feedback has pointed out important aspects that require additional clarity or information and helped us to improve our paper.

In the following, reviewers' comments are provided in blue, and our responses are in black. Changes to the manuscript made in response to the reviewer are in green.

**REVIEWER 1:**

In this manuscript, Che et al. run the latest generation of UKESM to investigate aerosol direct, semi-direct, and indirect effects from biomass burning smoke plumes produced by agricultural burning over the southeast Atlantic Ocean. The headline finding (in my read) is that the semi-direct effect of cooling due to increased cloudiness from a stronger cloud-top inversion dominates the overall radiative forcing, offset substantially by the direct effect of smoke absorption and reinforced marginally by indirect effects. The manuscript is well organized and the findings appear sound for the most part. (I do have some questions below, mainly pertaining to the Twomey effect). The text could use some areas of clarification and potentially additional information. I would not anticipate any major new analyses would need to be undertaken to address my comments, so I therefore recommend publication following minor revisions.

Thanks for the positive feedback! Significant reversions have been made to improve the clarity of the manuscript based on the comments from both reviewers.

**General comment:**

My biggest (really, only) concern with the results is that the indirect effect estimate seems unrealistically small given the change in cloud droplet number concentration you diagnose. From a simple back-of-the-envelope calculation of the Twomey effect, a doubling of cloud droplet concentration (you report 56% of CDNC are from biomass burning) should lead to a radiative forcing of O(10) W/m2, as in Lu et al. (2018). This is before taking into account the small liquid water increases you find. In my own work in the region (Diamond et al., 2020), I've estimated that a 5% increase in climatological CDNC from the influence of shipping produces a radiative forcing of ~-2 W/m2 during austral spring. I thus find it hard to believe that doubling CDNC would produce less than that in austral winter.

We thank the reviewer for pointing this out. This is actually a misunderstanding due to the inaccurate expression. The CDNC increased by the BBA is **up to** 56% only in some specific areas, not the average. We have calculated the mean contribution of BBA to CDNC during July and August in the SEA domain (the area this paper focuses on, ranging from  $30^{\circ}$  S to  $10^{\circ}$  N and from  $40^{\circ}$  W to  $30^{\circ}$  E), and the result shows the mean CDNC increase by BBA is around 13% and around 1.1% in the cloud box region. Although the indirect radiative cooling associated with the changes of CDNC is smaller than that in Diamond et al., (2020), some differences are to be expected taking significant model differences into account.

To avoid confusion, we have revised the manuscript as follows:

Though BBA can contribute up to 56% of total CDNC in some areas, its average contribution during July to August in the SEA is around 13%, much less than its contribution to the  $CCN_{0.2\%}$  budget fraction.

**Specific comments:**

1. Title: The title is currently rather non-informative. I would recommend highlighting that the semi-direct effects dominate (really the highlight of the paper in my opinion) in the title, and also mentioning that this is a global climate modeling study (as opposed to in situ or satellite observations or high-resolution process modeling).

Agree. Both reviewers have suggested to make the title more specific. According to the comments, the title has now changed to:

Cloud adjustments dominate the overall negative aerosol radiative effects of biomass burning aerosols in UKESM1 climate model simulations over the south-eastern Atlantic

2. Page 2, Line 1: I don't follow why the net sign of the aerosol radiative effect, rather than its magnitude, signifies the importance of aerosol in this region.

The net negative radiative effect of biomass burning aerosol results from the strong cooling of the semi-direct effect. Therefore, this sentence has been deleted, and the previous sentence has been revised as the following to underline the importance of the semi-direct effect.

Among the effects of biomass burning aerosols on the radiation balance, the semi-direct radiative effects (rapid adjustments induced by biomass burning aerosols radiative effects) have a dominant cooling impact over the SEA, which offset the warming direct radiative effect (radiative forcing from biomass burning aerosol-radiation interactions) and lead to overall net cooling radiative effect in the SEA. However, the magnitude and the sign of the semi-direct effects are sensitive to the relative location of biomass burning aerosols and clouds, reflecting the critical task of the accurate modelling of the biomass burning plume and clouds in this region.

3. Page 3, Line 3: I would argue that in situ results showing abundant biomass burning influence within the MBL at Ascension Island from the LASIC campaign (Zuidema et al., 2017) and throughout the SE Atlantic Klein-Hartmann box over the remote ocean from ORACLES (Diamond et al., 2018; Kacarab et al., 2020) are more directly relevant to your point about smoke-cloud interaction.

Thanks for the suggestion! We have adapted the comments and included the papers by Zuidema et al. (2018), Diamond et al. (2018), and Kacarab et al. (2020), in our manuscript.

However, recent studies found abundant biomass burning influence within the marine boundary layer (MBL) at Ascension Island from in-situ observations (Zuidema et al., 2018) and throughout the SEA from flight measurements (Diamond et al., 2018; Kacarab et al., 2020), confirming the interaction of BBA and clouds.

4. Page 3, Line 5: I don't believe the cited literature backs up the claim that "most" of the BBA is entrained into the MBL.

This sentence has been deleted, as the plume-cloud interaction has been described in the above text.

5. Page 3, Line 24: The large eddy simulation results of Yamaguchi et al. (2015) and Zhou et al. (2017) also seem relevant to cite/discuss here.

**Agree, the sentence has been revised as:**

Hence, related process studies mainly rely on high-resolution limited-area models (Gordon et al., 2018; Lu et al., 2018), as well as idealised large-eddy simulations (Yamaguchi et al., 2015; Zhou et al., 2017)

6. Page 4, Lines 1-2: The BBA plume subsiding and gradually meeting the rising MBL is true in the mean, but the picture is much more nuanced in reality, as instances of smoke-cloud contact were seen to be highly variable between and even with flights during the ORACLES and CLARIFY campaigns. It is also not necessarily the case that smoke-cloud contact corresponds instantaneously with the MBL being polluted, as discussed in Diamond et al. (2018).

Agree, we acknowledge that although the MBL is mostly polluted with BBA at Ascension Island (Zuidema et al., 2018), in other instances, the MBL is relatively clean, as influenced by the recirculation. However, around Ascension Island, both ground-based and flight observations have confirmed the frequently observed BBA (Zuidema et al., 2018, Wu et al., 2020). Therefore, it is reasonable to presume BBA have generally reached the MBL near Ascension Island. We also agree that even with the BBA entering the MBL; cloud properties are not affected by the instantaneous smoke-cloud contact, as discussed by Diamond et al. (2018). Therefore, to make it clearer, we have made the following revision:

These flight campaigns were carried out during the biomass burning seasons, and have provided an ideal dataset covering both BBA above and interacting with clouds, as previous studies have found that the BBA plume layer generally subsides and meets the gradually deepening marine boundary layer in the vicinity of Ascension Island and St Helena (Adebiyi et al., 2015). However, observations also indicate that the entrainment of BBA into the MBL can be intermittent, can require significant contact time (Diamond et al., 2018), and that recirculation patterns can result in clean MBL near Ascension Island.

7. Page 4, Line 13: Is dust included as one of the "five interactive log-normal aerosol modes"? I only count four other components (sulfate, sea salt, black carbon, organic carbon). The phrasing currently is confusing, as it sounds like the dust representation is entirely separate.

In this model configuration, dust is not included in the modal GLOMAP microphysics; it is treated separately in the model using a 6-bin externally mixed scheme(Woodward, 2001), while the interactive log-normal distribution simulates sulfate, sea salt, black carbon, and organic carbon. The "five interactive log-normal aerosol modes" in the manuscript refer to the modal aerosol modes (4 Soluble from nucleation to coarse and one insoluble of Aitken mode), not the aerosol species. The aerosol species are internally mixed in each mode. To avoid confusion, we have made a revision as below:

Aerosol and its interaction with clouds are represented by the UK Chemistry and Aerosol model (UKCA) (Mulcahy et al., 2020; O'Connor et al., 2014), including the modal aerosol

microphysics GLOMAP (Mann et al., 2010), with five interactive log-normal aerosol modes (four soluble modes from nucleation to coarse, and one insoluble Aitken mode) comprised of internally-mixed sulfate, sea salt, black carbon, and organic carbon. Mineral dust is represented separately by an externally mixed bin representation (Woodward, 2001).

8. Page 5, Line 9: Somewhere in the manuscript you should discuss the implications of only looking at one part of the biomass burning season (July-August). It is well known that the BB plume properties change over the course of the biomass burning season, influenced in part by meteorological shifts like the strengthening of the southern African Easterly Jet (AEJ-S) in September and October that corresponds with a more elevated plume (Adebiyi & Zuidema, 2016).

**Thanks for the suggestion. The corresponding discussion has been added as below:**

Two years are simulated in the model (2016 and 2017), however this analysis focuses on July and August, for consistency with the flight campaigns. Note although July and August can be used to represent BBA effects during the African fire season (July-October), this selection will also result in some uncertainties, as the BBA distribution and properties change over the course of the fire season, influenced in part by meteorological shifts, such as the strengthening of the southern African Easterly Jet (AEJ-S) in September and October, corresponding to a more elevated plume (Adebiyi and Zuidema, 2016).

9. Page 5, Lines 17-20: I'm surprised that you do not use any of the new products from MODIS or SEVIRI that account for above-cloud aerosol absorption. I would recommend trying the comparison using one of those products or at least discussing the issues with traditional AOD products that cannot retrieve AOD in the presence of clouds.

The primary model evaluation is done by comparing extinction measured from ORACLES and CLARIFY flights with the model. The MODIS AOD is used to validate the model bias in simulating the BBA plume after the evaluation. The model has been collocated with the MODIS AOD to reduce uncertainties (Schutgens et al., 2016). Although the AOD we used in the manuscript is the traditional product from MODIS, it is comparable with the model, after the collocation.

The main purpose of the AOD comparison is to evaluate the plume simulated in the model. The standard MODIS AOD retrieval is well evaluated and documented, although at Ascension Island, mean AOD (2001–2018) is slightly overestimated (around 0.02) by the MODIS (Gupta et al., 2020). However, the experimental nature of the above-cloud retrievals could be an issue, and require a very careful consistent determination of "above cloud", which is not always trivial in a climate model.

10. Figure 1: What altitude is being shown, or is this a column average? There is a large amount of vertical variability in the plume (as seen in Figure 2) so the 2D picture is a bit difficult to interpret.

This is the collocated model extinction, i.e., the extinction from the model is at the same time, latitude, longitude and altitude as the flight data. Thus, the comparison is point to point. To make it clearer, we have changed the figure caption as below:

Figure 1: Mean along track aerosol extinction coefficient [Mm-1] from the (a) UKESM1 model collocated to the flight tracks, (b) flight observations, and (c) differences between the model and observations. Note that the model extinction is under ambient conditions, whereas the measured extinction is for dry aerosols with relative humidity below 30%.

11. Page 6, Line 4: I do not understand why you only compare September AOD when the analysis focuses on July-August. As discussed earlier, there are known differences in plume location throughout the biomass burning season, in part driven by different meteorological factors between July-August and September-October. Thus, it's entirely possible that the model could represent one part of the season well but the other poorly if it is not representing those meteorological shifts properly. Figure S2 should be replaced with a new version including July and August.

We have changed the figure as illustrated below. The figure now shows the comparison of the mean AOD from model and MODIS (Terra and Aqua) from July to August, 2016-2017.

Figure S2. Mean (a) MODIS and (b) UKESM1 simulated AOD during July and August 2016-2017, and the (c) differences between MODIS and the model.

12. Page 6, Lines 12-13: As discussed above, although this description makes sense in the climatology, the picture we found in the field is much more complicated than the mean suggests. For one, much of the smoke in the marine boundary layer at a given location may have been entrained upstream and not necessarily reflect the properties of the plume above-cloud at the time of sampling (Diamond et al., 2018). It may be worth noting that although the mean field shows a plume subsiding from east to west, actual plume distribution and occurrence of plume-cloud contact at any given time is more nuanced.

This is similar to question 6. We agree that even when BBA enter MBL, cloud properties will not be affected by the instantaneous smoke-cloud contact as discussed by Diamond et al. (2018). Therefore, to make it clearer, we have made the following revision:

From east to west, the plume subsides and comes into contact with the clouds. At 5° W, the plume is generally inside the clouds, although the actual plume distribution and occurrence of plume-cloud contact at any given time can be more nuanced (Diamond et al., 2018). Thus, the BBA can interact and modulate cloud properties. This finding is also confirmed by previous studies (Adebiyi et al., 2015; Chand et al., 2009; Deaconu et al., 2019; Gordon et al., 2018).

13. Page 7, Lines 6-7: Are these percentages for the column burden? It may be worth also reporting the values for the marine boundary layer separately (if they differ), as the MBL CCN concentration is what matters most for cloud droplet activation.

Yes, these are percentages of the column burden CCN. We agree that the marine boundary layer CCN is more important in affecting cloud droplet number concentration. The CCN concentration and fraction are indeed quite different in MBL, we have a manuscript under preparation discussing the CCN source attribution, and find the mean number concentration of the BBA  $CCN_{0.2\%}$  during biomass burning season is ~75 cm-3 in the MBL and ~209 cm-3 in the plume layer. The BBA  $CCN_{0.2\%}$  fraction is ~40% in the MBL and ~84% in the plume layer, during the BB season. The detail of the CCN distribution will be discussed in the upcoming paper.

**14. Figure S3: Figure S3 is an exact copy of Figure 3. I believe the figure the authors meant to include would show the change in CCN due to BBA?**

Figure S3 shows the spatial and vertical distribution of total CCN, while figure 3 is the CCN from biomass burning. These two figures share the same pattern, indicating biomass burning is the main source of the CCN. We have changed figure S3 as below, to show the fraction of biomass burning CCN. Although the updated figure still has the same pattern as the CCN from BBA, which highlights the contribution of BBA to CCN.